# A Computation-Efficient Group Key Distribution Protocol Based on a New Secret Sharing Scheme

**Runhai Jiao [1,\*], Hong Ouyang [2], Yukun Lin [1], Yaoming Luo [3], Gang Li [4], Zaiyu Jiang [2] and Qian Zheng [2]**

[1] School of Control and Computer Engineering, North China Electric Power University, Beijing 102206, China; linyk95.gm@gmail.com

[2] China-Power Information Technology Co., Ltd., Beijing 100089, China; 13811915863@163.com (H.O.); jiangzaiyu@sgitg.sgcc.com.cn (Z.J.); zhengqian_bjtu@163.com (Q.Z.)

[3] State Grid Jiangxi Electric Power Corporation, Nanchang 330077, China; luoyaoming1979@163.com

[4] State Grid Chongqing Electric Power Co. Electric Power Research Institute, Chongqing 400022, China; 13883088031@sina.cn

\* Correspondence: runhaijiao@ncepu.edu.cn

**Abstract:** With the development of 5G and the Internet of Things (IoT), mobile terminals are widely used in various applications under multicast scenarios. However, due to the limited computation resources of mobile terminals, reducing the computation cost of members in group key distribution processes of dynamic groups has become an important issue. In this paper, we propose a computation-efficient group key distribution (CEGKD) protocol. First, an improved secret sharing scheme is proposed to construct faster encryption and decryption algorithms. Second, the tree structure of logical key hierarchy (LKH) is employed to implement a simple and effective key-numbering method. Theoretical analysis is given to prove that the proposed protocol meets forward security and backward security. In addition, the experiment results show that the computation cost of CEGKD on the member side is reduced by more than 85% compared with that of the LKH scheme.

**Keywords:** key distribution; multicast; secret sharing; logical key hierarchy

## 1. Introduction

Multicast is a one-to-many communication technology that can reduce the sender load and increase network-bandwidth utilization. With the development of 5G and the Internet of Things (IoT), application scenarios of multicast services are becoming increasingly abundant. Many emerging applications, such as network video conferences, Internet of vehicles, and other multicast applications, have higher requirements for bandwidth, continuity and real-time communication so the multicast technology is used to reduce the consumption of network bandwidth and improve efficiency [1–3]. To ensure the security of group communication, the group key shared by all legitimate group members is used to encrypt data in the communication channel [4]. Therefore, group key management is the core issue of multicast security. It is difficult to effectively manage the group key of large dynamic communication groups. Every time a member joins or leaves a communication group, the group key must be changed. On the one hand, when a member joins, the new member cannot obtain the previous group key, which is called backward security. On the other hand, when a member leaves, members still in the group must be able to efficiently calculate the new key, while the leaving member cannot obtain it, which is called forward security. Moreover, besides satisfying these security restrictions, it is also necessary to minimize the communication, computation, and storage costs of the rekeying process.

Group key management has been extensively and intensively studied. According to the introduction of the development of group key management in the literature [5,6], in the earliest

group key management protocol (GKMP), a central group controller (GC) directly unicast the rekeying message to each group member; thus, its computation and communication cost were linear with group size, resulting in poor scalability. Therefore, researchers have proposed hierarchical rekeying schemes [4,7–13] based on tree structure; the most typical scheme is the logical key hierarchy (LKH) scheme proposed by K.W. Chung et al. [4] and D. Wallner et al. [12]. It is a centralized group key management scheme that obviously reduces GC load. Tree structure is used to manage keys, where a leaf node represents a group member, the internal node connected to it represents its private key, and the other internal nodes are auxiliary keys, also known as key encryption keys. Each auxiliary key is shared by the group members connected to it for multicast, and the root node represents the group key. In the rekeying process, the LKH scheme uses a symmetric encryption algorithm and sends the rekeying message encrypted by the auxiliary key through multicast. The communication and computation cost are both $2 \log N$, where $N$ is the group size, which makes group key management for large-scale communication groups feasible. An improvement of LKH was proposed in the literature [14] in which a one-way function tree (OFT) is employed to improve the key-generation and rekeying process. The key of the leaf node is generated by GC, and the rest of the keys are calculated by a given formula. In the rekeying process, nearly half of the bandwidth consumption is reduced, but the computation cost of the group member to obtain the group key is increased.

Along with the extensive application of mobile terminals with limited computation capability, it is a critical problem to reduce the computation cost of the rekeying process. Scholars have proposed many lightweight protocols to balance security and efficiency. On the one hand, some protocols simplify the process of key distribution, or use dynamic routing and maximum distance separable code to distribute key update messages [15–19]. On the other hand, based on threshold secret sharing scheme [20,21], lightweight encryption and decryption algorithms are implemented to replace the traditional encryption algorithms used in LKH and OFT schemes [9,22–26]. Some polynomial-based multicast key distribution framework has also achieved good results in reducing the overhead of encryption and decryption algorithms [27,28].

## 1.1. Contributions

In this paper, inspired by threshold secret sharing, we propose a computation-efficient group key distribution (CEGKD) protocol based on a new secret sharing scheme. The main contributions of this paper can be summarized as follows:

- A new secret-sharing scheme is proposed. The corresponding polynomial degree of it is lower, which greatly simplifies the process of secret recovery. In addition, this scheme enables GC to construct an encrypted rekeying message when the secret shares of all authorized members have been determined, which avoids the transmission of secret shares.
- A simple and efficient node-coding method for a logical key tree is proposed and its update algorithm is given, which is the basis for implementing the new encryption and decryption scheme in the tree structure of LKH.
- A CEGKD that satisfies forward security and backward security is proposed, and its specific implementation is given. The number of polynomials to be constructed by GC is equal to the depth of the key tree, and the degree of polynomials is equal to the degree of the key tree.

## 1.2. Related Works

In general, most of the existing group-key distribution protocols focus on the balance between communication cost, computation cost, storage cost, and communication security, and cannot be applied well in the scenario where the terminal's computation capability is weak and the requirement for communication security is high. On the basis of LKH key update mechanism, using light encryption algorithm to distribute secret key can improve efficiency. Therefore, the key distribution schemes based on secret sharing, proposed by Shamir [20] and Blakley [21] in 1979, receive increasing attention [29],

since secret sharing has many good characteristics. For example, its security does not depend on any unproven assumptions, and the encryption and decryption algorithms are both very concise and practical. The LKH [22], JKKO [23], and STL [24] schemes are all based on secret sharing, but they can only distribute the group key once. The HL [25] scheme can distribute group keys multiple times, but its communication cost is linear with the group size. Harn and Lin [9] propose an authenticated group key transfer protocol (AGKTP) based on a secret-sharing scheme in which the members first share a secret with GC when they join a group. In the rekeying process, GC broadcasts the rekeying message to all group members, and only authorized group members can recover the group key. Unfortunately, the degree of polynomials in this scheme is equal to the number of group members, so the computational cost in large groups is enormous. Lih-Chyau Wuu et al. [26] proposed a group-key management scheme based on (2, 2) secret sharing that reduces the degree of polynomials to 1, but the number of polynomials to construct is equal to the number of members, which brings huge computation cost in large commutation groups. Therefore, a new secret-sharing scheme is needed to solve the key-distribution problem of large communication groups. Actually, the shamir threshold secret sharing scheme is a method of hiding information using polynomials. Ganesan V.C. et al. [27] and Mahmood Z. et al. [28] develop key distribution frameworks to distribute the group key using polynomial expressions without encryption for improving the efficiency in terms of communication, computation, and storage costs.

Based on the star topology, many scholars have proposed a lightweight key update mechanism other than LKH to improve the efficiency of key distribution. X.S. Li et al. [15] proposed a periodic batch rekeying method. In a given time period, GC collects join and leave requests of members, and finally updates the key tree only once. The batch processing method reduces communication and computation costs caused by frequent changes of group members but introduces security risks and reduces service quality. Instead of using symmetric encryption algorithms, L. Xu et al. [16] proposed a multicast key distribution scheme that used maximum-distance separable codes to reduce the computation cost of the key-generation process. Xiang L. [17] proposed a multipath routing scheme based on weak security-network coding, which can probabilistically guarantee the confidentiality of data transmissions and improve efficiency. These above schemes reduce the computation cost of multicast group members and improve communication efficiency, but the security of the group communication is sacrificed to some extent. Yoneyama K. et al. [18] propose a new provably secure two-round dynamic MKD (DMKD) protocol under the star topology with a central authentication server in which the join and leave phases need smaller computation and communication costs than the distribution phase. Chen Y.R. et al. [19] proposed KeyDer-GKM based on a semi-stateful rekey mechanism to reduce the number of decryption operations.

The rest of this paper is organized as follows. In Section 2, some preliminaries are given. Our proposed scheme is stated in Section 3. In Section 4, security and efficiency analysis is provided. In Section 5, we evaluate the performance of the proposed scheme. Finally, the paper is concluded in Section 6.

## 2. Preliminaries

### 2.1. Logical Key Hierarchy

LKH is a centralized group key management scheme based on a logical key tree, which is suitable for managing the group keys of large groups. This solution introduces a trusted, secure central GC to manage the group using the key tree. After a member is authenticated by GC, a private key of the member shared with GC is stored in the leaf node of the key tree. The auxiliary key stored by the internal node is shared with the connected members and is used to effectively and safely transfer the rekeying message by multicast when the group members change. The key stored in the root node is the group key (GK), also known as the traffic-encryption key. Each member holds all keys on the path from the leaf node representing itself to the root node, while the GC holds all the keys on the tree; such

a set of keys is called the key path. As shown in Figure 1, member $m_1$ holds $\{k_{1-9}, k_{123}, k_1\}$, where $k_{1-9}$ is the GK, $k_{123}$ is the auxiliary key shared by $m_1, m_2, m_3$, and $k_1$ is the private key of $m_1$.

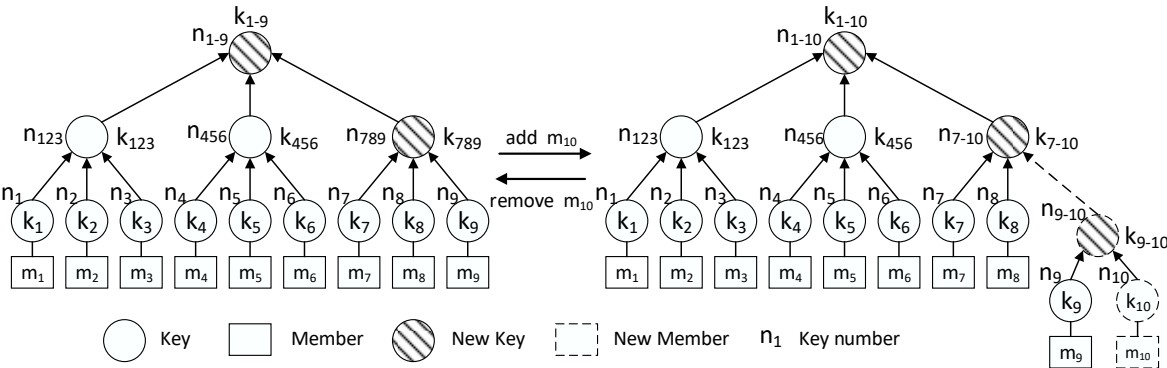

**Figure 1.** Key tree update process.

The joining or leaving of members can be divided into two parts: updating the key tree and distributing the new keys. Since only the GC holds the complete key tree, updating the key tree only needs to be performed by the GC. Three rekeying strategies are proposed in the literature [4]: User-Oriented, Key-Oriented, and Group-Oriented rekeying. The first strategy has a large computation cost, while the other two have a large communication cost. The Group-Oriented rekeying strategy was used below. Figure 1 shows the key-tree update process when the number of members changes between 9 to 10.

(1) Joining a key tree

When a member joins the group, the GC authenticates the membership and generates the personal key for the new member and inserts a corresponding node into the key tree. Then GC sends the key to the new member through a secure unicast channel. In order to ensure backward security, the GC updates all keys on the key path of the member in the logical key hierarchy and issues a rekeying message. For example, in Figure 1, new member $m_{10}$ requests to join, GC generates corresponding private key $k_{10}$ after verifying its identity, then updates all keys on the path from the parent node of $k_{10}$ to the root node ($k_{1-9}$ is updated as $k_{1-10}$, $k_{789}$ is updated as $k_{7-10}$), and, finally, GC issues the rekeying message:

$$S \rightarrow m_{10} : \{k_{1-10}, k_{7-10}\}_{k10}$$

$$S \rightarrow \{m_1, \cdots, m_9\} : \{k_{1-10}\}_{k1-9}, \{k_{7-10}\}_{k789}$$

where $\{k_{1-10}\}_{k1-9}$ represents $k_{1-10}$ encrypted with $k_{1-9}$.

The new member cannot calculate the previous group key $k_{1-9}$ according to the new group key $k_{1-10}$, thus ensuring the security of multicast communication content. This property is defined as backward security.

(2) Leaving a key tree

When a group member requests to leave, the GC deletes the corresponding leaf node in the key tree, updates all keys on the path from its parent node to the root node to ensure forward security, and finally issues a rekeying message. For example, in Figure 1, group member $m_{10}$ requests to leave, GC deletes node $k_{10}$ from the key tree, then updates $k_{1-10}$ and $k_{7-10}$ to $k_{1-9}$ and $k_{789}$, respectively, and finally the GC issues a rekeying message:

$$S \rightarrow \{m_1, \cdots, m_9\} : \{k_{789}\}_{k7}, \{k_{789}\}_{k8}, \{k_{1-9}\}_{k123}, \{k_{1-9}\}_{k456}, \{k_{1-9}\}_{k789}$$

After the new key is generated, the new key generated in the GC is securely sent to the required group members through key distribution. For a full $d$-ary tree T, if T contains N leaf nodes, the height

of the tree is $h = \log_d N$, GC stores $(dN-1)/(d-1)$ keys, and each member stores $\log_d N$ auxiliary keys. The communication and computation cost of the rekeying process are both $O(\log_d N)$. In the worst case, group members need to perform $\log_d N$ decryption operations. Compared with GKMP, the tree structure of LKH greatly reduces the communication and computation cost of the GC caused by the change of group members, but increases the storage cost of the GC and group members, and the computation cost of group members.

The leaving member cannot calculate the current group key $k_{1-9}$ according to the previous group key $k_{1-10}$. This property is defined as forward security.

*2.2. Shamir Threshold Secret-Sharing Scheme*

In the Shamir threshold secret sharing scheme, each member in the group $M(|M| = n)$ denoted as $m_i (1 \le i \le n)$ holds a secret share. A secret $s$ to be distributed can be recovered as long as any $t$ of the $n$ secret shares are obtained, and $t$ is called the threshold value. In the following, the secret distributor is the GC above, $q$ is a large prime number, and $GF(q)$ is the finite field of prime order $q$.

Secret division: First, the distributor GC randomly selects a polynomial $f(x) = \sum_{j=0}^{t-1} a_j x^j \mod q$ with a maximum degree of $t-1$ over $GF(q)$, Where $a_1, \cdots, a_j$ are random numbers on $GF(q)$ and $a_0 = s$. Second, the GC randomly selects n non-zero and mutually different numbers $x_i$, where $x_i \in GF(q)$, $(1 \le i \le n)$, and calculates $s_i = f(x_i) \mod q$ for all $i$. Then, the distributor GC secretly transmits $s_i$ to member $m_i$, which is the secret share of member $m_i$. Finally, GC publishes all $x_i$ as public parameters.

Secret recovery: Secret $s$ can be recovered by the cooperation of any $t$ members. A set of $t$ numbers is denoted as $B = \{1, 2, \ldots, t\}(|B| = t)$, and $f(x)$ is calculated from the Lagrange interpolation method [20]:

$$f(x) = \sum_{i \in B} C_{B_i}(x) \cdot s_i \mod q \tag{1}$$

where $C_{B_i}(x) = \prod_{j \in B \setminus \{i\}} \frac{x - x_j}{x_i - x_j} \mod q$, so the secret is:

$$s = f(0) = \sum_{i \in B} C_{B_i} \cdot s_i \mod q \tag{2}$$

where $C_{B_i} = \prod_{j \in B \setminus \{i\}} \frac{x_j}{x_j - x_i} \mod q$.

The Shamir threshold secret sharing scheme has the following good properties:

Security: When the number of shares obtained by an adversary are fewer than the threshold value $t$, the adversary cannot obtain any information about the secret. In addition, it is a scheme for Anticollusion Attacks

Idealism: Each secret share $s_i$ is the same size as the secret.

Scalability: When the threshold value is unchanged and new users join, the GC can calculate the corresponding new secret shares without affecting the existing secret share and distribute them to the new users.

## 3. Our Proposed Protocol CEGKD

In the LKH scheme, when the membership changes, the members who are still in the group need to update all keys on their key paths. In the case of a large communication group, group members need to perform decryption operations several times. Due to the limited computation resources of mobile terminals, if traditional symmetric encryption algorithms, such as AES, are used in the rekeying process, it takes too long for mobile terminals to decrypt the rekeying messages which lead to a long processing delay shown in Table 5 and they cannot participate in group communication in time. This is unacceptable for some time-sensitive communication requirements. Therefore, a faster encryption and decryption scheme is needed to reduce the computation cost of group members in decrypting the

rekeying messages. In the LKH scheme, each member holds all the keys in its key path, which can be regarded as secret sharing. This enables the GC to distribute the group key using the secret-sharing scheme. However, in the traditional threshold secret-sharing scheme, secret shares need to be generated by the secret distributor and transmitted to all encrypted members. It is meaningless to transmit other encrypted information in order to safely transmit the group key. In this section, we first propose a new form of secret-sharing scheme that enables the secret distributor to directly encrypt the message when the secret shares held by each member have been determined, and its decryption algorithm is simpler on the group-member side. Then, a simple and effective key-numbering method is implemented on the tree structure of LKH. Based on these, CEGKD is proposed, and the rekeying process is described in detail.

### 3.1. New Form of Secret Sharing Scheme

In this section, we give another form of a secret-sharing scheme, which enables the secret distributor to calculate the corresponding polynomial when the secret shares of all authorized members have been determined, so that any authorized member can obtain the secret. This avoids the computation and communication cost caused by encrypting and transmitting secret shares between GC and group members. In fact, if encrypted transmission is required between GC and group member, the transmission content can be directly set as the group key to be updated without any subsequent protocols. The polynomial degree is high if the threshold value is large, which leads to an excessive encryption and decryption cost. Therefore, a new secret sharing scheme with no extra transmission and a faster secret recovery algorithm is necessary.

A set of existing secret shares is denoted as S $= \{s_1, s_2, \ldots, s_t\}(|S| = t)$, and their number set is X $= \{x_1, \cdots, x_t\}(|X| = t)$.

**Secret division:** First, the distributor GC randomly selects a random number r, and then calculates $y_i = s + H(r, s_i)$ for each secret share $s_i$, where $H$ is a hash function, and obtains a series of different points $(x_1, y_1), \cdots, (x_1, y_t)$. Then, according to the Lagrange interpolation method, a polynomial with a maximum number of $t - 1$ is obtained on $GF(q)$:

$$f(x) = \sum_{i \in B} C_{B_i}(x) \cdot y_i \, mod \, q \tag{3}$$

where $C_{B_i}(x) = \prod_{j \in B \setminus \{i\}} \frac{x - x_j}{x_j - x_i} \, mod \, q$. After uniting like terms, $f(x)$ is converted into the following form:

$$f(x) = a_{t-1}x^{t-1} + \cdots + a_1 x + a_0 \tag{4}$$

Finally, GC publishes public parameters P $= \{r, coef\}$, where $coef = \{a_{t-1}, \cdots, a_0\}$.

**Secret recovery:** Any authorized member can recover the secret GC through its secret share $s_i$ and its number $n_i$, as well as public parameter P. After receiving the public parameters, members recover polynomial $f(x) = a_{t-1}x^{t-1} + \cdots + a_1 x + a_0$ according to $coef$, and calculate the secret as follows:

$$s = f(x_i) - H(r, s_i) \tag{5}$$

**Security analysis**

**Theorem 1.** *The authorized members can calculate the secret according to the parameters, and the calculation result is unique.*

**Proof.** There are $t$ points on the plane $(x_1, s + H(r, s_i)), \cdots, (x_t, s + H(r, s_t))$, according to Lagrange interpolation theorem [30], the only $t - 1$ degree polynomial passing through these $t$ points on the plane can be determined. For all $x_1, \cdots, x_t$, the difference $s$ between $f(x_i)$ and $H(r, s_i)$ is the same value

$s$. Therefore, all authorized members can calculate $f(x_i)$ and $H(r, s_i)$ in Equation (5) according to their corresponding number $x_i$ and the random number $r$ in the public parameters. □

**Theorem 2.** *The attacker cannot calculate secret using the public parameters provided by GC.*

**Proof.** In our scheme, the secret is hidden in the polynomial generated by Equation (3), and the adversary can only obtain public parameter P. The adversary can infer the number of secret shares through P and calculate $f(x_1), \cdots, f(x_t)$, but since the adversary does not know $s_1, \ldots, s_t$, it is impossible to calculate $H(r, s_1), \cdots, H(r, s_t)$, and thus, s cannot be calculated. That is, the enemy cannot calculate secret s without obtaining the secret share. □

**Theorem 3.** *The authorized members cannot calculate the secret shares of others through the obtained secret.*

**Proof.** Once the member obtains the secret s, the hash function value $H(r, s_i) = f(x_i) - s$ corresponding to any secret share $s_i$ can be calculated. Due to the unidirectionality of the hash function, it is impossible to find the value of $s_i$ according to the $H(r, s_i)$. □

    Actually, the two secret-sharing schemes are different applications of the Lagrange interpolation method, and they obviously have equivalent security. However, it is necessary to make such changes that enable the secret sharing scheme to be applied more efficiently in the encryption and decryption algorithm of the key distribution process. For the convenience of the following description, the encryption and decryption method are first defined before introducing the key-distribution protocol.

    **Encrypt (s, S, X)** The input of the encryption algorithm consists of a secret s, a secret share set S, and a set $X = \{x_1, \cdots, x_t\}$, and outputs a public parameter P.

    **Decrypt (r, *roe*, $x_i$ $s_i$)** The input of the decryption algorithm consists of public parameter P and secret share $s_i$, owned by the decrypter and its corresponding number $x_i$, and the secret s is output.

### 3.2. Group-Key Distribution Protocol

    In this paper, an LKH tree structure is used to manage keys. Based on the secret-sharing scheme proposed above, the CEGKD is proposed. Taking the key represented by the child node of the new key as the secret share, GC encrypts the new key to form a rekeying message, and group members decrypt the rekeying message, layer by layer, using the key they have mastered. The encryption and decryption method adopt the new form of the secret-sharing scheme proposed in the previous section, and the protocol is described below by taking the trigeminal tree as an example.

### 3.2.1. Key-Numbering Rule

    Since the key number is required in the encryption and decryption process of the secret-sharing scheme, and the number does not need to be kept secret, a simple but effective recursive key-numbering rule using string concatenation is proposed:

1. Number the group key as $(01)_2$
2. For other keys k, assuming that the parent node number of k is n, the number of $k$ is $d\|n$, where $\|$ represents string concatenation, and d is a binary string. If k is the first child of its parent node, then $d = (01)_2$; if k is the second child, then $d = (10)_2$; if k is the third child, then $d = (11)_2$.

    For example, in the ternary key tree shown in Figure 1, the key number of $k_1$ is $n_1 = (010101)_2$, the key number of $k_5$ is $n_5 = (101001)_2$, and the key number of $k_{789}$ is $n_{789} = (1101)_2$.

### 3.2.2. Joining a Group

(1)  Apply to join

After the new member applies to join, the GC verifies the group member's identity, generates the group member's personal key and leaf node *leaf*, then finds the internal node *insert* with the shallowest depth and the number of its child nodes less than 3 from the key tree, and inserts *leaf* into the tree as the child node of ***insert***. If the key tree is a full trigeminal tree, the splitting operation as shown in Figure 1 is performed to generate a new tree node $k_{9-10}$, and then $k_{10}$ is inserted into the tree as a child node of $k_{9-10}$. Finally, GC sends the personal key to the new member through a secure channel.

(2)  Update keys and send rekeying message

The GC updates all keys on the path from the parent node of the joining node to the root node, from bottom to top, and each new key corresponds to a key update polynomial on $GF(q)$:

$$f(x) = a_{t-1}x^{t-1} + \cdots + a_1 x + a_0$$

The construction of the polynomial method is as follows: denote the updated node as ***node***, the key to update is $k'$, the node has $t$ children, and the corresponding numbers and keys are $(n_1, k_1), \cdots, (n_t, k_t)$; then, the GC takes $k'$ as the secret, sets $S = \{k_1, \cdots, k_t\}$ as the secret share set, sets $\mathbf{X} = \{n_1, \cdots, n_t\}$, and calculates public parameters $P = \textbf{Encrypt}(k', \mathbf{K}, \mathbf{X})$. P corresponds to key update polynomial $f(x) = a_{t-1}x^{t-1} + \cdots + a_1 x + a_0$.

Assuming that h keys are updated after the group members join, the GC constructs and multicasts a rekeying message to the group members:

$$\text{RM} = \{r, coef_1, \cdots, coef_h, SN\} \tag{6}$$

where $coef_i$ is the coefficient of key update polynomial $f_i(x)$ and $SN$ is the update serial number. If the key tree is split when the new node joins, $SN$ is the key number of the parent node of the new member; otherwise, $SN$ is the number of the joining node.

(3)  Obtain updated key

Suppose a member has a total of $l$ keys $km_1, \cdots, km_l$, $km$ arranged in the order from the root node to the leaf node, i.e., $km_1$ is the group key GK and $km_l$ is the member's private key. Private key $km_1, \cdots, km_l$ corresponds to key number $n_1 \cdots, n_l$. After receiving RM, group members update their stored keys as follows:

1.  Align private key number $n_l$ to the right of $SN$. If $n_l = SN$, and it is not the new member, update the private key number to $n_{l+1} = 01\|n$, and set $km_{l+1} = km_l$.
2.  From right to left, compare $n_l$ and the SN of each two bits to obtain the same number in succession. Once the difference is found, the process is terminated. Denote the result as $i$. For the new member, the result is set to $i - 1$.
3.  Calculate new key $km_i$ according to the following formula:

$$km_i = \textbf{Decrypt}(r, roe_i, n_i, km_{i+1}) \tag{7}$$

4.  Replace the $i$-th key stored by the member with $km_i$.
5.  Reduce the value of I by 1 and repeat Steps 2 to 5 until $km_1$ is obtained.

For example, in Figure 1, member $m_{10}$ requests to join the group communication. The GC updates $k_9, k_{789}, k_{1-9}$ to $k_{9-10}, k_{7-10}, k_{1-10}$, respectively, and then calculates:

$$P_3 = \textbf{Encrypt}(k_{9-10}, \{k_9, k_{10}\}, \{n_9, n_{10}\})$$

$$P_2 = \textbf{Encrypt}(k_{7-10}, \{k_7, k_8, k_{9-10}\}, \{n_7, n_8, n_{9-10}\})$$

$$P_1 = \textbf{Encrypt}\big(k_{1-10}, \{k_{123}, k_{456}, k_{7-10}\}, \{n_{123}, n_{456}, n_{7-10}\}\big)$$

Extract $coef_1, coef_2, coef_3$ from $P_1, P_2, P_3$, and broadcast rekeying message

$$RM = \big\{r, coef_1, coef_2, coef_3, (111101)_2\big\}$$

After receiving the rekeying message, group member m9 updates the private key number to $n_9 = (01111101)_2$, and then calculates, in turn:

$$k_{9-10} = \textbf{Decrypt}(r, coef_3,\ n_9\ k_9)$$

$$k_{7-10} = \textbf{Decrypt}(r, coef_2,\ n_{9-10}\ k_{9-10})$$

$$k_{1-10} = \textbf{Decrypt}(r, coef_1,\ n_{7-10}\ k_{7-10})$$

Finally, replace $k_{1-9}, k_{789}$ with $k_{1-10}, k_{7-10}$.

### 3.2.3. Leaving a Group

(1) Send a leave message

When a member leaves a communication group, the member sends their private key number to the GC. The GC determines the corresponding node on the key tree and then deletes it. Delete the node as *leaf 1* and its parent node as **insert**. If only one child node *leaf 2* is left in the **insert** after deletion, node **insert** is deleted, and *leaf 2* is used to replace its position in the key tree.

(2) Update key and send a rekeying message

If no key tree-node replacement occurs when members leave, all keys on the path from the leaving node to the root node are updated; if a key tree-node replacement occurs, all keys on the path from the parent node of the replacement node to the root node are updated. In the process of updating the key from bottom to top, each new key corresponds to key-updating polynomial $f(x) = a_{t-1}x^{t-1} + \cdots + a_1 x + a_0$ over $GF(q)$, and its construction method is the same as that in the previous section. Next, assuming that h keys are updated after group members join, the GC constructs and multicasts a rekeying message to the group members:

$$RM = \{r, coef_1, \cdots, coef_h, SN\}$$

If key tree-node replacement does not occur when members leave, SN is the key number of the leaving node; otherwise, SN is the number of the parent node of the replaced node.

(3) Get updated key

After receiving the RM, group members update their stored keys in the same way as above.

For example, in Figure 1, after member $m_{10}$ leaves, the GC deletes $k_{10}$ and replaces node $k_{9-10}$, with $k_9$, updates $k_{7-10}, k_{1-10}$ to $k_{789}$, and $k_{1-9}$, respectively, and then calculates:

$$P_2 = \textbf{Encrypt}(k_{789}, \{k_7, k_8, k_9\}, \{n_7, n_8, n_9\})$$

$$P_1 = \textbf{Encrypt}\big(k_{1-9}, \{k_{123}, k_{456}, k_{789}\}, \{n_{123}, n_{456}, n_{789}\}\big)$$

Extract $coef_1, coef_2$ from $P_1$ and $P_2$, and then broadcast rekeying message

$$RM = \big\{r, coef_1, coef_2, (1101)_2\big\}$$

After receiving the rekeying message, group member m9 updates the private key number to $n_9 = (111101)_2$, deletes $k_{9-10}$ and $n_{9-10}$, and then compares $n_9$ and SN $= (1101)_2$ to obtain $i = 2$; then, it is calculated in turn:

$$k_{789} = \mathbf{Decrypt}(r, coef_2, n_9\ k_9)$$

$$k_{1-9} = \mathbf{Decrypt}(r, coef_1, n_{789}\ k_{789})$$

Finally, replace $k_{1-10}$, $k_{7-10}$ with $k_{1-9}$, $k_{789}$.

## 4. Security and Efficiency Analysis

### 4.1. Security Analysis

The Shamir threshold secret-sharing scheme is the most practical and effective secret-sharing scheme. It has been tested in practice for many years since it was proposed. In Section 3.1, we proved that, in the new form of the secret-sharing scheme, the adversary cannot decrypt secret s without obtaining the secret share. CEGKD applies a new secret-sharing scheme to construct and decrypt the rekeying message. In this process, random number $r$, polynomial coefficient $coef$, and key number SN are all public, and only the keys held by members are confidential. The security of the proposed CEGKD is analyzed in terms of forward and backward security.

(1)  Forward security

Taking the case shown in Figure 1 as an example, GC updates $k_{1-10}$, $k_{7-10}$ to $k_{1-9}$, and $k_{789}$, respectively, after $m_{10}$ leaves the communication group. $k_{1-9}$ and $k_{789}$ can be obtained by the following method:

$$k_{789} = \mathbf{Decrypt}(r, coef_2, n_7\ k_7)$$

$$k_{789} = \mathbf{Decrypt}(r, coef_2, n_8\ k_8)$$

$$k_{789} = \mathbf{Decrypt}(r, coef_2, n_9\ k_9)$$

$$k_{1-9} = \mathbf{Decrypt}(r, coef_1, n_{123}\ k_{123})$$

$$k_{1-9} = \mathbf{Decrypt}(r, coef_1, n_{456}\ k_{456})$$

$$k_{1-9} = \mathbf{Decrypt}(r, coef_1, n_{789}\ k_{789})$$

$m_{10}$ holds the following information:

1.  $r$ and $coef_1, coef_2$ obtained from RM;
2.  key and key number on the path before leaving; and
3.  all key numbers inferred from key number rules.

As $m_{10}$ does not have any secret shares in $k_7, k_8, k_9, k_{123}, k_{456}, k_{789}$, it cannot execute any of the above six-decrypt algorithm. Therefore, $m_{10}$ cannot calculate group key GK $= k_{1-9}$ after leaving and cannot decrypt the contents of the group communication after leaving. Therefore, CEGKD satisfies forward security.

(2)  Backward security

Taking the case shown in Figure 1 as an example, GC updates $k_{1-9}$, $k_{789}$ to $k_{1-10}$, and $k_{7-10}$, respectively, after new member $m_{10}$ joins the communication group. $k_{1-9}$ and $k_{789}$ can be obtained by the following method:

$$k_{789} = \mathbf{Decrypt}(r, coef_2, n_7\ k_7)$$

$$k_{789} = \mathbf{Decrypt}(r, coef_2, n_8\ k_8)$$

$$k_{789} = \mathbf{Decrypt}(r, coef_2, n_9\ k_9)$$

$$k_{1-9} = \textbf{Decrypt}(r, coef_1,\ n_{123}\ k_{123})$$

$$k_{1-9} = \textbf{Decrypt}(r, coef_1,\ n_{456}\ k_{456})$$

$$k_{1-9} = \textbf{Decrypt}(r, coef_1,\ n_{789}\ k_{789})$$

$m_{10}$ holds the following information:

1. $r$ and $coef_1, coef_2$ obtained from RM;
2. key and key number on the path before leaving; and
3. all key numbers inferred from key number rules.

As m10 does not have any secret shares in $k_7$, $k_8$, $k_9$, $k_{123}$, $k_{456}$, $k_{789}$, it cannot execute any of the above six-decrypt algorithm. Therefore, $m_{10}$ cannot calculate group key GK = $k_{1-9}$ before joining and cannot decrypt the contents of the group communication before encryption. Therefore, CEGKD satisfies backward security.

*4.2. Efficiency Analysis*

In this section, we evaluate the performance of the proposed CEGKD from three aspects: computation cost, communication cost, and storage cost. Denote the current number of members for group communication as N. In order to compare with the AES algorithm with 128 bits of key length in LKH, the key and key number were set to be 16 and 4 bytes, respectively.

4.2.1. Computation Cost

In this section, we evaluate the computation cost of the GC and group members in two ways. Method 1 evaluates computation cost by the number of encryption and decryption algorithms required for a rekeying process. Considering that the basic operation in the encryption and decryption algorithm in this scheme is a modular operation over $GF(q)$, which is quite different from the time required for a basic operation in AES, Method 2 evaluates the computation cost by the number of basic operations over $GF(q)$ required for the encryption and decryption algorithm.

**Method 1.** The GC constructs the RM through encryption algorithm **Encrypt($k'$,K,X)**. For a full $d$-ary tree with N leaf nodes, $\log_d N$ nonprivate keys are updated when new members join, so the GC needs to execute $\log_d N$ **Encrypt($k'$,K,X)**. After receiving the RM, the group member executes decryption algorithm **Decrypt(r, *roe*, $x_i\ s_i$)** to obtain the updated key, and needs to execute $\log_d N$ times **Decrypt(r, *roe*, $x_i\ s_i$)** at most. Similarly, for a full $d$-ary tree with N leaf nodes, when the group member leaves, the number of times the GC needs to perform the encryption calculation is $\log_d N - 1$, and group members need to perform a decryption calculation at most $\log_d N - 1$. The computation cost calculated by Method 1 is shown in Table 1.

**Table 1.** Computation cost of computation-efficient group-key distribution (CEGKD) under Method 1.

| Cost | Member Join | Member Left |
|---|---|---|
| Group controller (GC) | $\log_d N$ | $\log_d N - 1$ |
| Group member | $\log_d N$ | $\log_d N - 1$ |

**Method 2.** Operations include modular addition, modular subtraction, modular multiplication, modular division, and hash operations. When there are d secret shares, that is, $|K| = |X| = d$, the GC needs to construct a polynomial with the maximum order not exceeding $d - 1$. From Equation (3), it can be seen that $d \cdot 2^{d-1}$ modulo addition, $d(d-1)$ modulo subtraction, $d(d-3)2^{d-2} + d^2 - d$ modulo multiplication, $d$ modulo division, and $d$ hash are required in the process of calculating polynomial coefficients and merging similar terms. The GC needs to execute $\log_d N$ times the encryption algorithm, where $|X| = 2$ in 1 encryption algorithm, and $|X| = d$ in $\log_d N - 1$ encryption algorithm. The number

of operations required by the GC are shown in Table 2. From Equation (5), it can be seen that, in the decryption algorithm $|X| = d$, since group members know all node numbers of their paths, they can perform a preoperation on node number $n^2$, so only $d - 1$ modulo addition operation, 1 modulo subtraction operation, $d - 1$ modulo multiplication operation, 0 modulo division operation, and 1 hash operation are required. The maximum number of operations required by group members is shown in Table 2.

**Table 2.** CEGKD computation cost under Method 2.

| Operations | GC | Group Member |
|---|---|---|
| Modular addition | $d(\log_d N - 1) \cdot 2^{d-1} + 4$ | $(d-1)(\log_d N - 1) + 1$ |
| Modular subtraction | $d(d-1)(\log_d N - 1) + 2$ | $\log_d N$ |
| Modular multiplication | $d(\log_d N - 1)(d - 1 + (d-3)2^{d-2})$ | $(d-1)(\log_d N - 1) + 1$ |
| Modular division | $d(\log_d N - 1) + 2$ | $0$ |
| Hash operations | $d(\log_d N - 1) + 2$ | $\log_d N$ |

Because the basic operation of a symmetric encryption algorithm is different from this scheme, computation cost cannot be directly compared. In the next section, we compare the computation cost of CEGKD with that of other schemes through simulation experiments.

### 4.2.2. Communication Cost

In this section, we use the number of bytes occupied by RM to evaluate the communication cost. When the group-communication members change, the GC updates the keys and multicasts RM to all group members, so the communication cost is the size of the RM. Assuming that h keys are updated, the rekeying message multicast by the GC to the group members is:

$$RM = \{r, coef_1, \cdots, coef_h, SN\} \tag{8}$$

For a full $d$-ary tree with N leaf nodes, when a new member joins, the GC needs to construct $\log_d N$ polynomials, where the number of coefficients of one polynomial is 2, and the remaining $\log_d N - 1$. The polynomial contains $d$ coefficients, so there are $(d-1)(\log_d N - 1) + 2$ coefficients in the RM. When the group members leave, the GC needs to construct $\log_d N - 1$ polynomials, and each contains d coefficients, so there are $d(\log_d N - 1)$ coefficients in the RM. Each coefficient size is 16 bytes. In addition, the random number $r$ is 16 bytes, and key number $SN$ is 4 bytes. Therefore, the communication overhead of this solution is shown in Table 3.

**Table 3.** CEGKD communication cost.

| Cost | Member Join | Member Left |
|---|---|---|
| | $16((d-1)(\log_d N - 1) + 2) + 20$ | $16d(\log_d N - 1) + 20$ |

Therefore, the communication cost of this scheme is the same as the communication cost of the LKH scheme [4].

### 4.2.3. Storage Cost

In this section, we evaluate the storage cost of the GC and group members based on the number of bytes of keys and the key numbers stored by the GC and group members.

A full $d$-ary tree with N leaf nodes has a total of $\frac{dN-1}{d-1}$ nodes, corresponding to $\frac{dN-1}{d-1}$ keys and key numbers stored by the GC. The depth of the key tree is $\log_d N$, so that there are $\log_d N$ nodes in the path from the leaf node to the root node, corresponding to $\log_d N$ keys and key numbers stored by group members. In order to speed up the calculation, group members also need to store the precalculation

results of each power of the key number. Therefore, the storage overhead of this scheme is shown in Table 4.

**Table 4.** CEGKD storage cost.

| Storage cost | GC | Group Member |
|---|---|---|
| | $20 \cdot \frac{dN-1}{d-1}$ | $(16 + 4(d-1)) \log_d N$ |

The storage costs of the GC and group members in the LKH scheme are $16(dN-1)/(d-1)$ and $16 \log_d N$, respectively. Therefore, the storage costs of the GC and group members in this scheme are greater than those in LKH scheme [4].

## 5. Experiments

In this section, we compare the performance of CEGKD with the LKH scheme and the others through experiments. The performance of the LKH scheme reached the best [4] when the degree of the key tree was 4. Therefore, the key-tree degree used in the LKH scheme was set to 4, an AES algorithm with a 128 secret key length was used, and ECB codebook encryption mode was adopted. For the balance of security and efficiency, SHA1 was chosen as the secure one-way hash function in this scheme; in addition, R is a 128-bit strong random number, and q is a 130-bit secure prime number. The degree of the key tree used by the CEGKD scheme was tested under various conditions. The experiments were carried out on a core i7 2.30 GHz machine with a 12 Gbyte memory running Linux Redhat 9.0.

### 5.1. Computation Cost

In this section, we compare and analyze the computation costs of CEGKD and other group-key distribution protocols. Results are shown in Figures 2 and 3.

(1)  GC computation cost. In addition to the computation cost analyzed in the Section 4, GC computation cost also comes from operations such as generating key tree nodes and traversing the key tree. As can be seen from Figure 2, except that the computation cost of PMUKD scheme is constant, the cost of other methods is logarithmic to the group size. Under the same key-tree degree, the computation cost of the LKH scheme is close to that of the CEGKD scheme. For the CEGKD scheme, the bigger the degree of the key tree, the higher the GC computation cost is.

(2)  Computation cost of group members. After receiving the rekeying message, the group members in the CEGKD scheme obtain the updated key by modulo operation and hash operation over finite field $GF(q)$, which is faster than the AES decryption algorithm in the LKH scheme. In addition, group members can precalculate the powers of numbers in polynomials, which also makes the CEGKD scheme more efficient on the group-member side. As can be seen from Figure 3, although the computation costs of all schemes except PMUKD are logarithmically related to the number of group members; the computation cost of the CEGKD scheme on the group-member side is much lower than that of the LKH scheme and the others'. The larger the communication-group size is, the greater the computation-efficiency advantage of CEGKD at the member side. Although the computation cost of PMUKD scheme is constant, the computation cost of CEGKD on the group member side is still the smallest among all schemes within the foreseeable range of group size. For the CEGKD scheme, the bigger the degree of the key tree is, the lower the computation cost of the group members.

The computation time of the rekeying process compared to other key-distribution schemes proposed in References [16,19,26,27] was as shown in Table 5, for a selected multicast-group size. Since the number of encryption basic operations of various protocols except PMUKD is logarithmic to the group size, and the operation time of basic operations is independent of the group size, results under other multicast-group sizes were similar.

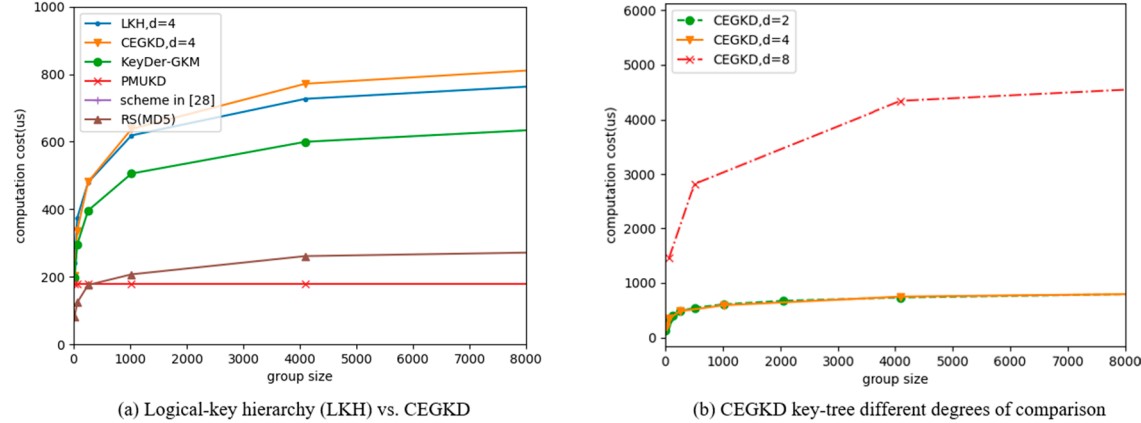

**Figure 2.** GC computation cost.

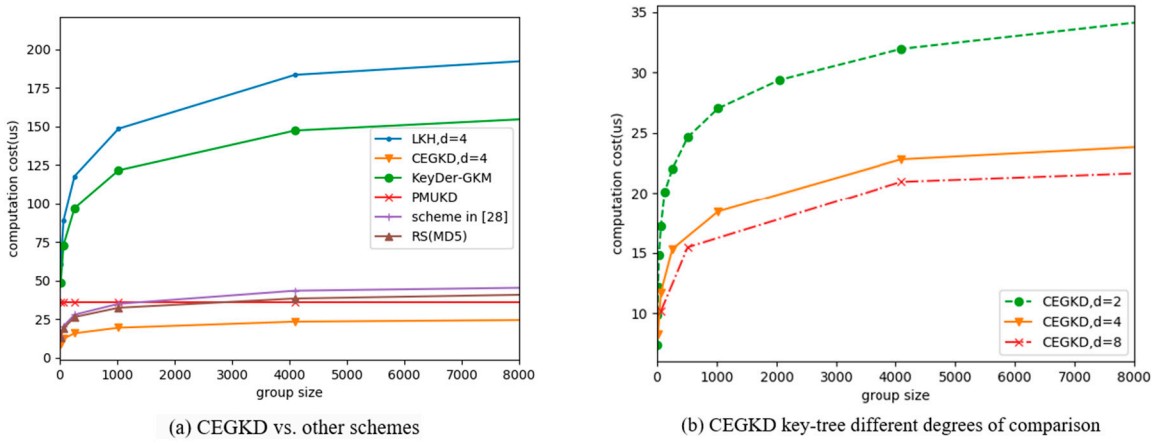

**Figure 3.** Group-member computation cost.

**Table 5.** Computation time compared to other schemes (multicast-group size of 8192).

| Time (μs) | CEGKD | PMUKD [27] | RS(MD5) [16] | Scheme in [26] | KeyDer-GKM [19] | LKH [4] |
|---|---|---|---|---|---|---|
| GC | 894 | 178 | 294 | 14963 | 706 | 840 |
| member | 26 | 36 | 45 | 49 | 170 | 211 |

*5.2. Communication Cost*

In this part, we compare and analyze the communication cost of LKH and CEGKD and count the RM size when a group member joins or leaves the group. Results are shown in Figure 4.

As can be seen from Figure 4, the communication cost is logarithmically related to the number of group members. In the case where the number of group members is the same, the greater the number of key trees is, the greater the communication cost of the CEGKD scheme. When the key tree degrees are the same, the communication cost complexity of the two schemes is $O(\log_d N)$, but the actual communication cost of the LKH scheme is slightly smaller than the communication overhead of the CEGKD scheme. This is because CEGKD's rekeying message contains a random number R with a size of 16 bytes.

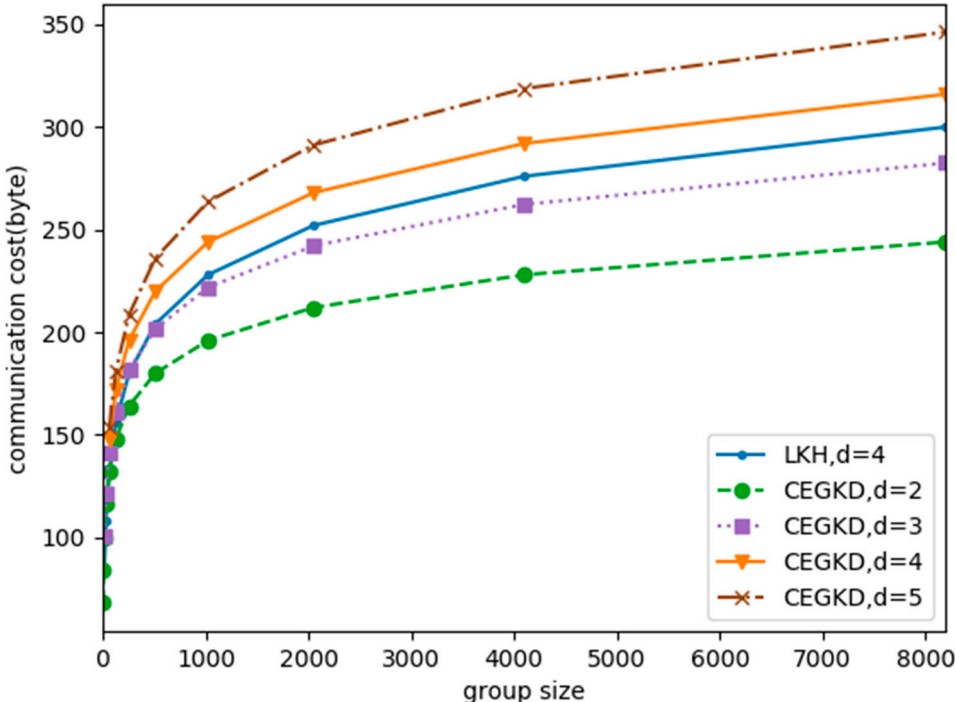

**Figure 4.** CEGKD and LKH communication cost.

## 5.3. Storage Cost

In this section, we compare the storage cost of LKH and CEGKD, and compare the storage cost of the GC and the members when a group member joins or leaves the group. Results are shown in Figure 5.

(1)  GC-side storage cost. As can be seen from Figure 5, as the size of the communication group increases, storage-cost increases of the two schemes are both linear with member size. GC storage cost under the CEGKD scheme is slightly larger than the value under the LKH scheme. For the CEGKD scheme, the greater the degree of the key tree is, the smaller the GC storage overhead.

(2)  Group-member-side storage cost. As can be seen from Figure 5, the storage costs of the two schemes are both logarithmically related to member size. The storage cost of group members under the CEGKD scheme is greater than that of LKH scheme because the precomputation results of the powers of the key numbers are both to be stored.

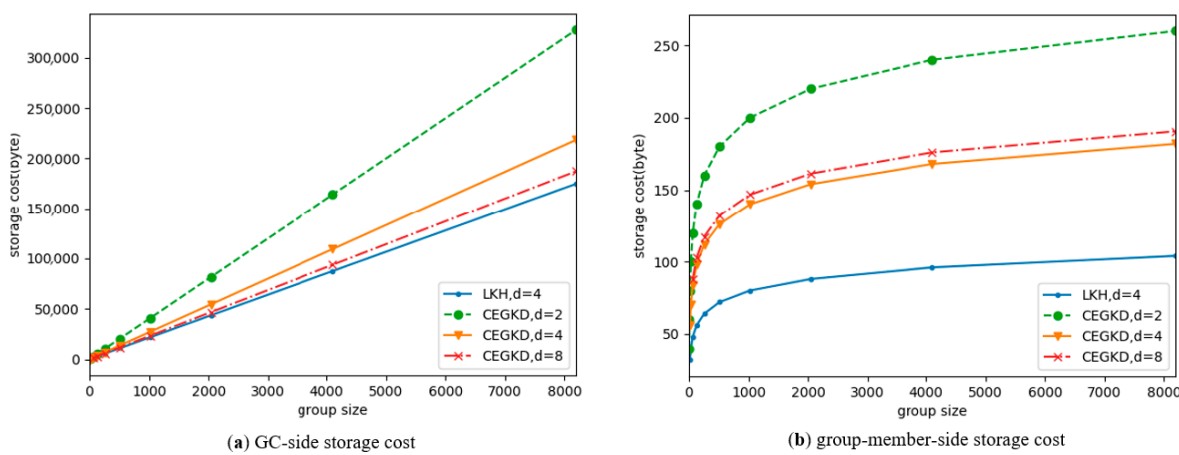

**Figure 5.** Storage cost of CEGKD and LKH.

*5.4. Analysis*

Considering the computation cost, communication cost, and storage cost of the GC and group members, CEGKD has the best performance when the key-tree degree is 4, which is the same as the LKH. When the degrees of the key tree in CEGKD and LKH are both 4, the cost of CEGKD compared with LKH is shown in Table 6.

**Table 6.** Cost of CEGKD compared with LKH.

| Change | Computation Cost | Communication Cost | Storage Cost |
|--------|------------------|--------------------|--------------|
| GC | Increase by about 3% | Increase by 16 bytes | Increase by 25% |
| Member | Decrease by more than 87.7% | | Increase by 75% |

Experiment results show that CEGKD can effectively reduce the computation cost of group members. At the same time, its communication cost and storage cost are both close to the LKH scheme. Although the storage cost on the group-member side increases by 75%, this is due to its small base. In fact, when the group size reaches 8192, the storage cost on the CEGKD member side is only 182 bytes. However, compared with other schemes, CEGKD has a higher computation cost on the GC side. In particular, the computation cost of PMUKD is a small constant on both GC and group members sides, which has great advantages in large-scale group communication.

Therefore, considering all kinds of situations, PMUKD is a better choice when the performance of communication group manager is insufficient. CEGKD is a good choice when the terminal's computation resource is extremely limited.

## 6. Conclusions

To solve the problem of insufficient computation resource of mobile terminals in the key distribution process, based on the LKH scheme, we propose a computation-efficient group-key distribution protocol based on a new secret-sharing scheme. Firstly, by designing a new secret sharing scheme, an efficient encryption and decryption algorithm is proposed. The degree of the corresponding polynomial is reduced in the proposed scheme. Moreover, precalculating makes the CEGKD scheme more efficient on the group-member side. On the premise of ensuring security, the efficiency of decrypting rekeying messages by group members was improved. Then, in the case of managing the key using the tree structure of the LKH, a simple and effective key-numbering method was realized. According to the experiment results, our proposed protocol could significantly reduce the computation cost of group members in the rekeying process. The wireless terminal may occasionally disconnect in group communication. In the future, we will study the key distribution scheme in which group members can obtain the latest group key in the case of missing key update messages in the middle period. The safety of the proposed scheme is still analyzed with various attacks.

**Author Contributions:** Y.L. conceived of the presented idea, developed the theory, implemented the algorithms and proposed CEGKD. R.J. and H.O. performed the simulations and the tests; R.J., Y.L., G.L., Z.J and Q.Z. verified the simulation and test results and supervised the whole research procedure of this paper.

**Funding:** This work was supported in part by the project "Research on the Application of the Integration Model Based on IEC-CIM in the Distribution Network and the key Technology of Service Channel" of State Grid Corporation of China, and the Fundamental Research Funds for the Central Universities (2018ZD06).

**Conflicts of Interest:** The authors declare no conflict of interest.

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
