# Peer review of "A Computation-Efficient Group Key Distribution Protocol Based on a New Secret Sharing Scheme"

_information, doi:10.3390/info10050175_

Reviewer 1 Report

In the CEGKD protocol of this paper, the authors first propose an improved secret sharing scheme to construct faster encryption and decryption algorithms. Secondly, the authors use LKH to implement a simple and effective key numbering method. Research on this aspect is worthy of recognition. However, there seem to be some problems in this, and the authors are suggested to give an explanation or explanation.

1.      It is suggested that the first part be divided into two parts, the introduction and the related work, in order to more clearly understand the ideas of this paper and the research progress of the existing works.

2.      In Section 3.1, the authors only give a simple security analysis. However, in order to be more convincing, the authors are advised to give a proof of security based on a difficult problem.

3.      In the comparative analysis of the experiment, the group size reached 8000. In reality, there are not so many terminals in a certain range, and the conclusion is not real enough. Please give a more granular comparison. In addition, it is recommended that in the comparison chart, different schemes should be processed in different colors to facilitate differentiation.

4.      In Section 5, the authors only simply compare their scheme to LKH. In addition, the references are relatively old, and it is recommended to add new schemes for comparison to show the paper's innovation and persuasiveness.

5.     In the conclusion, it is best to give a prospective part of the future work.

Author Response

Response to Reviewer 1 Comments

 Thank you very much for your great efforts on our manuscript. We also appreciate the your valuable suggestions and questions. In the process of answering and solving these questions, our research has been further promoted.

Point 1: It is suggested that the first part be divided into two parts, the introduction and the related work, in order to more clearly understand the ideas of this paper and the research progress of the existing works. 

 Response 1:

We have divided the first part into two parts, and roughly classified the previous methods to facilitate readers to sort out the research context. At the end of introduction (lines 64-70) we have added a general description of the relevant work.

In the related works section, we divided the research into two types. On the one hand, some protocols simplify the process of key distribution, or use dynamic routing and maximum distance separable code to distribute key update messages. We have added two papers after 2016 in this direction (line 121-125). On the other hand, based on threshold secret sharing scheme, lightweight encryption and decryption algorithms are implemented to replace the traditional encryption algorithms used in LKH and OFT schemes. Moreover,in the second part, besides the encryption improvement scheme based on threshold secret sharing scheme, we also consider the secret hiding method based on polynomial (line 106-109) .

Point 2: In Section 3.1, the authors only give a simple security analysis. However, in order to be more convincing, the authors are advised to give a proof of security based on a difficult problem.

 Response 2: We proposed three theorems and proved in section 3.1.(line 267-284).Theorem 1. The authorized members can calculate the secret according to the parameters, and the calculation result is unique. Theorem 2. The attacker cannot calculate secret using the public parameters provided by GC. Theorem 3. The authorized members cannot calculate the secret shares of others through the obtained secret.

Point 3: In the comparative analysis of the experiment, the group size reached 8000. In reality, there are not so many terminals in a certain range, and the conclusion is not real enough. Please give a more granular comparison. In addition, it is recommended that in the comparison chart, different schemes should be processed in different colors to facilitate differentiation.

 Response 3: Thank you very much for this suggestion. In the fifth section of the paper, we conducted a supplementary experiment to show the calculation cost of each scheme when the group size is 16 to 8000, and to distinguish the curves of different schemes with different colors. (including KerDER-GKM [19], proposed in 2016; PMUKD [30], proposed in 2016, scheme in [28] proposed in 2014 and RS(MD5) [16] proposed in 2008). In different pictures, the color of the uniform scheme curve is the same. And when the degree of the key tree is other values, CEGKD is represented by dotted lines. Moreover, we revised the conclusion of the experiment and pointed out the shortcomings of the proposed scheme (line 500-502/512-520/525-530/569-574).

At present, the vast majority of group communication scale is indeed difficult to reach 8000. However, under some special applications, this scale is possible. During the implementation of the supporting project of the paper, we use high-frequency intelligent electronic device clusters to detect electricity theft in a community, and multicast is used between devices to collect and exchange data required by machine learning algorithms. Since the detection range of the measurement equipment will change with the scheduling of the algorithm, the members of the communication group will change at a certain frequency. Although this is only a pilot project, the number of our intelligent electronic devices has reached thousands. In many fields of multicast communication, the scale of communication groups is increasing rapidly. Therefore, it is necessary to study a key update protocol with higher key distribution efficiency under the condition of larger scale than traditional communication groups. In fact, a study published on IEEE Transactions on Parallel and Distributed Systems as early as 2008 estimated a group size of tens of thousands. In addition, since the number of encryption basic operations of various protocols except PMUKD is logarithmic to the group size, and the operation time of basic operations is independent of the group size, results under other multicast-group sizes were similar.

 Point 4: In Section 5, the authors only simply compare their scheme to LKH. In addition, the references are relatively old, and it is recommended to add new schemes for comparison to show the paper's innovation and persuasiveness.

 Response 4: We have added two papers after 2016 in this direction (line 121-125/106-109). And in section 5.1, a comparative experiment is added to make the experimental results clearer. (including KerDER-GKM [19], proposed in 2016; PMUKD [30], proposed in 2016, scheme in [28] proposed in 2014 and RS(MD5) [16] proposed in 2008)

 Point 5: In the conclusion, it is best to give a prospective part of the future work.

 Response 5: We rewrote the conclusion and added a prospective part of the future work in line 590-603.

 Thank you again for your valuable advice!

Reviewer 2 Report

In general, it's a qualified paper with fine structures, methodologies, analyses and detailed simulation on the proposed methodology. Authors propose a CEGKD group key management protocol to improve the efficiency of encryption and decryption algorithms. Also, this study uses a tree based LKH to implement a key numbering method. Eventually, authors compare CEGKD with LKH mechanisms and show that CEGKD has the better computation cost. Several fine ideas are introduced on this study, I therefore suggest this paper can be accepted. I also encourage authors can compare more related methods with this proposed scheme.

Author Response

Response to Reviewer 2 Comments

Thank you very much for your careful reading of our paper and your affirmation of our research. In the future, we will study the key distribution scheme in which group members can obtain the latest group key in the case of missing key update messages in the middle period. The safety of the proposed scheme is still analyzed with various attacks.

Thank you again for your affirmation of our research!

Reviewer 3 Report

Thanks for the submission. It sounds interesting, but is partly hard to ready if no deep pre-knowledge is in place. Further I would recommend to do some updates to improve readability, especially shorten the length of some sentences (e.g., lines 363-368) and do not write in such a complicate way.

Comments:

line 13/27 introduce the acronam IoT for Internet od Things

line 29: what do you understand under "Internet of vehicles"? Car-to-Car communication --> VANETs

line 31/46/125: I guess [1] is a wrong reference here.

line 46: reference needs to be placed behind "et al." and also the right first author should be named.

line 87: what do you mean by "(2,2)"?

line 101: Here you mention the first time "forward and backward security", but throughout the paper you never mention what stands behind. And thus, it is also hard in the evaluation to understand your prove of support.

Figure 1: it is unsharp and misses legend

line 136: How is the identity verified?

line 141: how does the operation "update all keys" work? In case you describe it in the paper make a cross refrence to the corresponding section.

line 154: what is the relation between t and n? If not mentioned, it is hard to understand what you mean by "any t of n"

line 169: I believe the properties you list here come frome a reference. So you need to add it here.

line 177: you gave your solution a name, so add the acronym here in the section title

line 182: what do you understand under "takes too long". This is a grading and if you do not give hints about what you mean with this, it is hard to grade it.

line 225: "Lagrange interpolation method" needs a formal refernce

Figure 2/4: what is the difference to Figure 1 except for "n"?

Throughout Section 3/4: why are parts written in bold?

line 387: Formula 5 --> Equation 5, stay with the same wording throughout the paper

line 428: ECB stands for what?

Figure 5/6/7: you already use (a) and (b) under the indifigual figures, so place the titel there as well and not included in the overall figure title.

Section 6 reads more as a summary than a conclusion.

References are incomplete as publishers and months are missing. as well use long terms of journals/conferences and not the acronyms.

Improve the evaluation part, because it is really hard to follow. Try to add more cross references between different sections so the reader can see the links more easily.

Author Response

Response to Reviewer 3 Comments

Thank you very much for your great efforts on our manuscript. In particular, thank you for your detailed comments, which are of great significance to improve our research.

Point 1: line 13/27 introduce the acronam IoT for Internet od Things 

Response 1: The acronym IoT for Internet of Things has been introduced (line 13/27).

Point 2: line 29: what do you understand under "Internet of vehicles"? Car-to-Car communication --> VANETs.

Response 2: We believe that Car-to-Car communication and VANETs are different stages of development of the same thing. In the past Car-to-Car communication model, the car was only regarded as a communication terminal to obtain information. In today's VANET application, some approaches could include the use of deep learning and cognitive computing along with network link data sharing through Big Data technologies. With the application of new communication technology and artificial intelligence technology in VANET, VANET  application, such as automobiles, have higher requirements for bandwidth, continuity and real-time communication(added in line 31-32). Researchers have proposed many new communication network models, and we have added two references about IoV and other IoT applications (line 32/614/616, references 2 and 3). On the other hand, we hope to improve the encryption and decryption protocol for these applications, so we propose CEGKD.

Point 3: line 31/46/125: I guess [1] is a wrong reference here.

Response 3: We are very sorry for the wrong reference order. [1] should be revised to [2]. As we added references, the reference orders in lines 33, 48 and 164 were updated to 4.

Point 4: line 46: reference needs to be placed behind "et al." and also the right first author should be named.

Response 4: Reference has been placed behind "et al." in line 48. The first author of Reference 12 is Wallner, D.M., and we corrected this error in line 636.

Point 5: line 87: what do you mean by "(2,2)"?

Response 5: In the Shamir threshold secret sharing scheme, a secret can be recovered as long as any t of n secret shares are obtained. N is the total number of secret shares, and t is the secret recovery threshold. So the shamir secret sharing scheme is also called (t, n)-secret sharing. The reference Group Key Management based on (2, 2) Secret Sharing in line102 uses shamir threshold secret sharing scheme for key distribution, in which the total number of secret shares and the threshold value are both 2. Since this is a widely used term, some papers even put it directly on the title, so there is no special explanation in the paper.

Point 6: line 101: Here you mention the first time "forward and backward security", but throughout the paper you never mention what stands behind. And thus, it is also hard in the evaluation to understand your prove of support.

Response 6: For backward security and forward security, we give their definitions in lines 37/179-181 and 39/196-197 respectively.

Point 7: Figure 1: it is unsharp and misses legend

Response 7: We redrawn Figure 1 and added the legend.

Point 8: line 136: How is the identity verified?

Response 8: Digital signature can be used for identity verifying. This paper mainly studies the fast group key redistribution protocol. Certification application based on digital signature is relatively mature at present. We believe that readers interested in this paper have a basic understanding about digital signature and identity authentication. Besides, identity authentication is not closely related to the core content of the article, which does not affect the understanding of the article. Therefore, we have not carried out a detailed explanation.

Point 9: line 141: how does the operation "update all keys" work? In case you describe it in the paper make a cross refrence to the corresponding section.

Response 9: All key nodes are stored in a logical key hierarchy, usually a tree structure. The update operation is that the server calculates the new key and updates the storage content of the corresponding node in the tree. We believe that it is not difficult for readers to realize this in engineering, so this paper does not give a specific description of its realization. Actually, we believe that focusing on key redistribution algorithms and protocols is beneficial to readers' understanding.

Point 10: line 154: what is the relation between t and n? If not mentioned, it is hard to understand what you mean by "any t of n"

Response 10: The t secret shares are part of the total n secret shares. We adjusted the descriptions in lines 199-203 of the paper to make the relationship between t and n more clear.

Point 11: line 169: I believe the properties you list here come frome a reference. So you need to add it here.

Response 11: We have added the source of the formula in line 213.

Point 12: line 177: you gave your solution a name, so add the acronym here in the section title

Response 12: We added the acronym CEGKD in line 224.

Point 13: line 182: what do you understand under "takes too long". This is a grading and if you do not give hints about what you mean with this, it is hard to grade it.

Response 13: We explained in line 229-230 that it would lead to a long procesing delay and added a cross reference.

Point 14: line 225: "Lagrange interpolation method" needs a formal refernce

Response 14: We added the reference in line 270/673.

Point 15: Figure 2/4: what is the difference to Figure 1 except for "n"?

Response 15: The difference between fig. 1 and figs. 2-4 is that there is no splitting operation in fig. 1. At the beginning, we only want to show LKH in a simpler state for easy understanding. Now we have modified fig. 1 so that subsequent operations can be understood on fig. 1 and figs. 2-4 are deleted.

Point 16: Throughout Section 3/4: why are parts written in bold?

Response 16: There are many pages in this paper, so we want to help readers grasp the key points of the key update process in this way. Secret division and secret recovery are the two key processes of the proposed secret sharing scheme. Encrypt (s, S, X) and Decrypt (r,roe, x_i s_i) will be mentioned frequently below, so they are all written in bold. In the following, the corresponding parameters are also bolded to indicate that the variable will be used in Encrypt and Decrypt.

Point 17: line 387: Formula 5 --> Equation 5, stay with the same wording throughout the paper

Response 17: Formula on line 443/448 has been modified to Equation.

Point 18: line 428: ECB stands for what?

Response 18: ECB is the electronic codebook mode of AES algorithm, which is a parameter option when calling AES algorithm toolkit. It is explained in this paper to facilitate researchers to reproduce the implementation results.

Point 19: Figure 5/6/7: you already use (a) and (b) under the indifigual figures, so place the titel there as well and not included in the overall figure title.

Response 19: As some pictures were deleted, we updated the picture number. Title have been placed under the individual figures in fugure 2/3/5 .

Point 20: Section 6 reads more as a summary than a conclusion.

Response 20: We rewrote the conclusion and added a prospective part of the future work in line 590-603.

Point 21: References are incomplete as publishers and months are missing. as well use long terms of journals/conferences and not the acronyms.

Response 21: We changed the abbreviation of the journal name to full name on line 620/622/628/631/635/641/646/654/660/672.

Point 22: Improve the evaluation part, because it is really hard to follow. Try to add more cross references between different sections so the reader can see the links more easily.

Response 22: In section 3.1,We proposed three theorems and proved them in lines 267-284 to help readers understand the security of the algorithm.In section 5,we conducted a supplementary experiment to show the calculation cost of each scheme when the group size is 16 to 8000, and to distinguish the curves of different schemes with different colors. (including KerDER-GKM [19], proposed in 2016; PMUKD [30], proposed in 2016, scheme in [28] proposed in 2014 and RS(MD5) [16] proposed in 2008). In different pictures, the color of the uniform scheme curve is the same. And when the degree of the key tree is other values, CEGKD is represented by dotted lines. Moreover, we revised the conclusion of the experiment and pointed out the shortcomings of the proposed scheme (line 500-502/512-520/525-530/569-574).

Thank you again for your valuable advice!

Round  2

Reviewer 3 Report

The revised Version is fine for me, so i would it accept it the way it is.